# Thymoquinone Enhances Apoptosis of K562 Chronic Myeloid Leukemia Cells through Hypomethylation of *SHP-1* and Inhibition of JAK/STAT Signaling Pathway

**DOI:** 10.3390/ph16060884

**Published:** 2023-06-15

**Authors:** Futoon Abedrabbu Al-Rawashde, Ola M. Al-Sanabra, Moath Alqaraleh, Ahmad Q. Jaradat, Abdullah Saleh Al-Wajeeh, Muhammad Farid Johan, Wan Rohani Wan Taib, Imilia Ismail, Hamid Ali Nagi Al-Jamal

**Affiliations:** 1Department of Anatomy and Histology, Faculty of Medicine, Mutah University, Al-Karak 61710, Jordan; futoonrawashdeh1001@gmail.com; 2Department of Medical Laboratory Sciences, Faculty of Science, Al-Balqa Applied University, Al-Salt 19117, Jordan; ola.sanabra@bau.edu.Jo; 3Pharmacological and Diagnostic Research Center (PDRC), Faculty of Pharmacy, Al-Ahliyya Amman University, Amman 19328, Jordan; m.alqaraleh@ammanu.edu.jo; 4Department of Medical Laboratory Sciences, Faculty of Allied Medical Sciences, Mutah University, Al-Karak 61710, Jordan; ahmad.jaradat@mutah.edu.jo; 5Anti-Doping-Lab Qatar, Doha 27775, Qatar; a_alwajeeh@yahoo.com; 6Department of Haematology, School of Medical Sciences, Universiti Sains Malaysia, Kubang Kerian 16150, Malaysia; faridjohan@usm.my; 7School of Biomedicine, Faculty of Health Sciences, Universiti Sultan Zainal Abidin (UniSZA), Kuala Terengganu 21300, Malaysia; wanrohani@unisza.edu.my (W.R.W.T.); imilia@unisza.edu.my (I.I.)

**Keywords:** thymoquinone, CML, hypomethylation, *SHP-1*, JAK/STAT, *DNMT1*, *DNMT3A*, *DNMT3B*, *TET2*, *WT1*

## Abstract

The epigenetic silencing of tumor suppressor genes (TSGs) is critical in the development of chronic myeloid leukemia (CML). *SHP-1* functions as a TSG and negatively regulates JAK/STAT signaling. Enhancement of *SHP-1* expression by demethylation provides molecular targets for the treatment of various cancers. Thymoquinone (TQ), a constituent of *Nigella sativa* seeds, has shown anti-cancer activities in various cancers. However, TQs effect on methylation is not fully clear. Therefore, the aim of this study is to assess TQs ability to enhance the expression of *SHP-1* through modifying DNA methylation in K562 CML cells. The activities of TQ on cell cycle progression and apoptosis were evaluated using a fluorometric-red cell cycle assay and Annexin V-FITC/PI, respectively. The methylation status of *SHP-1* was studied by pyrosequencing analysis. The expression of *SHP-1*, *TET2*, *WT1*, *DNMT1*, *DNMT3A*, and *DNMT3B* was determined using RT-qPCR. The protein phosphorylation of STAT3, STAT5, and JAK2 was assessed using Jess Western analysis. TQ significantly downregulated the *DNMT1* gene, *DNMT3A* gene, and *DNMT3B* gene and upregulated the *WT1* gene and *TET2* gene. This led to hypomethylation and restoration of *SHP-1* expression, resulting in inhibition of JAK/STAT signaling, induction of apoptosis, and cell cycle arrest. The observed findings imply that TQ promotes apoptosis and cell cycle arrest in CML cells by inhibiting JAK/STAT signaling via restoration of the expression of JAK/STAT-negative regulator genes.

## 1. Introduction

Although the BCR-ABL oncogene is the main driver of CML, cumulative evidence indicates that epigenetic alterations such as aberrant DNA methylation is essential for CML development [1,2,3]. Targeting BCR-ABL with tyrosine kinase inhibitors (TKIs) such as imatinib (IM) has shown a response in CML patients. However, most CML patients do not recover and show drug resistance after receiving long-term treatment [4]. Therefore, new therapeutic approaches are needed to target aberrant signaling pathways, which are critical in CML development.

CML is associated with frequent mutations of genes involved in regulating the DNA methylation process, such as DNA-methyltransferases (DNMTs) mutations, including the *DNMT1* gene, *DNMT3A* gene, and *DNMT3B* gene mutations [5], and DNA demethylases mutations, including Wilms Tumor 1 (*WT1)* mutation and Ten Eleven Translocation 2 (*TET2*) mutation [5,6].

Reduced expression of tumor suppressor genes (TSGs) due to epigenetic silencing is crucial in developing CML and response to therapy [3,7]. The Src homology-1-domain containing protein-tyrosine phosphatase (*SHP-1*) functions as a TSG, which acts as a negative regulator of the Janus-tyrosine kinase/signal transducer and activator of transcription (JAK/STAT) pathway [3,7,8].

Epigenetic silencing of *SHP-1* due to promoter hypermethylation is known to cause JAK/STAT signaling hyperactivation in hematological malignancies, including leukemia [3,9]. Hyperactivation of STAT3 and STAT5 proteins enhances cell proliferation and inhibits apoptosis by modulating the expression of various genes involved in cellular proliferation, differentiation, and apoptosis [10]. Demethylating agents such as 5-Azacytidine (5-Aza) increase *SHP-1* gene expression, which inhibits JAK2/STAT3/STAT5 signaling [7,11]. However, DNA hypomethylating drugs have insufficient therapeutic efficiency [12,13]. Therefore, alternative therapies that effectively target aberrant DNA methylation, as well as tyrosine kinases, are vitally needed.

Phytochemical agents have safe and efficient activities for cancer treatment [14,15,16,17]. The phytochemical agents can target and control gene expressions by affecting both epigenetic status and genetic status [18]. Thymoquinone (TQ) is an important active compound extracted from *Nigella sativa* seeds [19]. TQ showed anti-cancer activities by modulating several cellular mechanisms essential for cancer development, such as modifying the epigenetic features in cancer cells [19,20].

TQ has been reported to inhibit cancer cell growth and enhance apoptosis in a diversity of cancers, including Glioblastoma multiforme cells [21], oral squamous cell carcinoma [22], colorectal and breast cancers [23], acute lymphoblastic leukemia [24,25], and myeloid leukemia [26,27]. Furthermore, TQ has been found to inhibit the activation of several signaling molecules in JAK/STAT and PI3K/Akt/mTOR pathways, which are directly involved in cancer cell proliferation, angiogenesis, and metastasis [28,29]. However, the ability of TQ to alter DNA methylation and its anti-leukemia activities are still not fully clear.

The current study investigated TQs anti-leukemic activities on K562 CML cells. Therefore, we identified DNA methylation of the JAK/STAT-negative regulator gene, *SHP-1*, to be the molecular mechanism that mediates TQs anti-leukemic activities.

The hypothesis of this study is that *SHP-1*, a negative regulator gene of JAK/STAT signaling, loses its tumor-suppressive activity in CML due to epigenetic silencing, leading to JAK/STAT signaling hyperactivation, and TQ can re-express this TSG via hypomethylation of the *SHP-1* promoter region. Therefore, the influence of TQ on *SHP-1* methylation and expression status was investigated. TQs effect on the *TET2* gene, *WT1* gene, *DNMT1* gene, *DNMT3A* gene, and *DNMT3B* gene expression was also evaluated. The ability of TQ to inhibit JAK/STAT signaling and their consequences on cell activities were also assessed in *BCR-ABL* positive K562 CML cells.

## 2. Results

### 2.1. Thymoquinone Enhances Apoptosis in K562 CML Cells

TQs effect on the rate of apoptosis of K562 CML cells was measured by using the Annexin V- FITC/PI determination of apoptosis method. K562 cells were exposed to 23, 15, and 11 μM TQ for 24, 48, and 72 h, sequentially. As shown in Figure 1A,B, the apoptotic cells were increased in a dose and time-dependent manner. The total apoptotic cells were only 12% following 24 h of exposure to 11 M of TQ, compared to 54% after 23 μM TQ for the same incubation period. Furthermore, the highest apoptosis rate (91%) was achieved following 72 h exposure to 23 μM compared to 54% after 24 h at the same dose.

### 2.2. Thymoquinone Delays Cell Cycle Progression in K562 Cells

Analysis of cell cycle progression was conducted to further determine TQs effects on K562 cells. Treatment of K562 cells was performed by exposing the cells to the IC_50_s; 23, 15, and 11 μM for a period of 24, 48, and 72 h, sequentially. After treatment for 24 h and 48 h, flow cytometry results revealed that TQ caused a significant elevation in the number of cells at the G2/M phase and reduced the number of cells at the G0/G1 phase in a time- and concentration-dependent manner. TQ prompted the highest accumulation at the G2/M phase (42.84%) after 48 h of 23 μM treatment compared to 26.56% in untreated cells. Furthermore, the percentages of cells in the S and G0 phases were not significantly changed (Figure 2A). After 72 h of treatment, TQ significantly increased the G2/M and G0 populations while it decreased the G0/G1 population. The most significant increase of the cells in the G0 phase (9.23%) was after treatment with 23 μM TQ compared to 2.91% in untreated cells (Figure 2A,B).

### 2.3. Thymoquinone Develops a Balanced EXPRESSION of the Genes That Regulate DNA Methylation in K562 CML Cells

To investigate TQs effect on the expression of the genes that regulate methylation of DNA (the *TET2* gene, *WT1* gene, *DNMT1* gene, *DNMT3A* gene, and *DNMT3B* gene in K562 cells), RT-qPCR analysis was performed for the TQ treated (15 μM for 48 h exposure) and control cells. TQ caused a significant reduction in *DNMT1* gene expression (1.6-fold, *p* = 0.002), *DNMT3A* gene expression (2.5-fold, *p* < 0.001), and *DNMT3B* gene expression (3.8-fold, *p* < 0.001) in comparison to the control. TQ caused a significant elevation in *TET2* gene expression (3.5-fold, *p* < 0.001) and *WT1* gene expression (2.7-fold, *p* < 0.001) in comparison to the control (Figure 3).

### 2.4. Thymoquinone Causes Hypomethylation of the SHP-1 Promoter in K562 CML Cells

Pyrosequencing was applied to study the influence of TQ on DNA methylation in K562 cells. The methylation of 4 CpG-sites located at the promoter 2 area of *SHP-1* was studied in TQ-treated-K562 cells and untreated-K562 cells. Pyrosequencing results demonstrated a significant reduction of methylation levels in the CpG sites after TQ exposure (81.2%) in comparison to 92% before treatment (*p* = 0.029) (Figure 4, Table 1).

### 2.5. Thymoquinone Enhances SHP-1 Expression in K562 Cells

The RT-qPCR method was applied to evaluate *SHP-1* expression in K562 cells. The results showed a significant increase in SHP-1 mRNA levels after 48 h of 15 μM TQ treatment. TQ caused an increase (2.7-fold) in *SHP-1* expression, which was statistically significant (*p* < 0.001) in comparison to the untreated cells (Figure 3).

### 2.6. Thymoquinone Induces JAK/STAT Signaling Inhibition in K562 Cells

TQs effect on the levels of expression for the STAT3 protein, p-STAT3 protein, STAT5 protein, p-STAT5 protein, JAK2 protein, and p-JAK2 protein in K562 cells was assessed via Jess-simple Western analyses. A concentration of 15 μM TQ was used to treat the cells for 48 h. The results demonstrated a significant impact by TQ, which decreased STAT3 protein intensity (*p <* 0.05), STAT5 protein (*p <* 0.001), as well as JAK2 protein (*p <* 0.05). TQ also potentially lowered the phosphorylation of STAT3 protein (*p <* 0.001), STAT5 protein (*p =* 0.029), along with JAK2 protein (*p* < 0.001) (Figure 5 and Table 2).

## 3. Discussion

Epigenetic silencing of TSGs plays a critical role in the pathogenesis of CML, which may cause resistance to TKIs [1,4,7]. Epigenetic silencing of *SHP-1* results in JAK/STAT pathway hyperactivation leading to uncontrolled proliferation and differentiation of the myeloid cells [9,30]. Moreover, restoration of the TSGs expression by DNA demethylation provides significant molecular targets in the treatment of leukemia. Conventional DNA hypomethylation has limited therapeutic efficacy [12,13]. Thus, new therapies which effectively reverse the epigenetic aberration of TSGs are vitally needed.

TQ has been shown to exert antitumor effects in several cancers [27,31,32]. However, the potential anti-leukemia effect of TQ through DNA demethylation of TSGs has not been investigated. In this study, the anti-leukemia activities of TQ on CML were investigated.

Regulation of DNA methylation by DNMTs plays a vital role in normal hematopoiesis [33]. *WT1*, along with *TET2*, plays an essential role in mediating the DNA demethylation process by recycling 5-methylcytosine to cytosine [34,35]. Aberrant DNA methylation mediated by mutations of *TET2*, *WT1*, *DNMT1*, *DNMT3A*, and *DNMT3B* genes is correlated to the pathogenesis of myeloid leukemia [33,34]. A balanced expression of TET2, WT1, and DNMTs is necessary for regulating the process of DNA methylation [5]. In this research, we studied TQs influence on *TET2*, *WT1*, *DNMT1*, *DNMT3A*, and *DNMT3B* gene expression in K562 cells. According to the findings, TQ significantly lowered DNMT3A, DNMT3B, and DNMT1 mRNA levels in K562 cells (Figure 3). Consistent with these findings, TQ exposure induced a significant decrease in *DNMT3A* and *DNMT1* expression in MV4-11 acute myeloid leukemia (AML) cells and the primary AML blast cells [27,36]. TQ has also reduced *DNMT3A, DNMT3B,* and *DNMT1* expression in acute lymphoblastic leukemia cells (Jurkat cells) [37,38].

Our findings also showed that TQ increased *TET2* and *WT1* gene expressions in K562 cells, as shown in Figure 3. Similarly, previously conducted research reported that TQ had increased the expression of *TET2* in Jurkat cells and in HECV human vascular endothelial cells [39] and increased *TET2* and *WT1* gene expressions in MV4-11 AML cells [36]. However, *WT1* gene expression was downregulated in TQ-treated HL60 AML cells [40], which might be attributed to the dual functioning of *WT1*, acting as an oncogene and TSG [41,42].

The JAK/STAT-negative regulator, *SHP-1*, is a TSG that inhibits the growth and metastasis of tumor cells [43]. Epigenetic silencing of *SHP-1* has been identified in hematological malignancies and leukemia cells, including CML cells [3]. Interestingly, in our previous studies, TQ induced *SHP-1* hypomethylation and enhanced *SHP-1* expression in MV4 -11 AML cells which led to growth inhibition and apoptosis enhancement [36], implying that TQ could be an effective anti-leukemia drug by targeting the epigenetically silenced genes that negatively regulate JAK/STAT in leukemia cells. In this study, the methylation status of *SHP-1* was examined in K562 cells to further understand TQs DNA hypomethylating activity in CML.

TQ has been reported to lower DNA-methylation of MV4-11 AML cells and inhibit cell growth [27,36]. Furthermore, TQ has been shown to enhance the expression of various TSGs that have been found to be silenced in several cancers, such as leukemia [37]. TQ has also been reported to increase the methylation level and decrease the mRNA level of the *TWIST1* oncogene, which resulted in growth inhibition of SiHa and CaSki cervical cancer cells [20], and BT549 human breast cancer cells [44]. In agreement with these findings, the findings of this study indicated that TQ lowered *SHP-1* promoter methylation (81.2%) in comparison to the untreated control cells (92%) (*p* = 0.029), as illustrated in Table 1 and Figure 4.

In the present research, we investigated the potential impact of TQ-induced hypomethylation on *SHP-1* gene expression in K562 cells. The findings demonstrated that TQ induced a significant elevation in SHP-1 mRNA levels (Figure 3). These results agree with a previous study in which 5-Azacytidine enhanced *SHP-1* expression in leukemia cells [45].

Silencing of *SHP-1* due to hypermethylation leads to constitutive activation of JAK/STAT signaling in myeloid leukemia [30]. In our investigations of K562 cells, we also evaluated the possible impact of *SHP-1* re-expression on the activation status of STAT5, JAK2, and STAT3. The data indicated that TQ induced a significant reduction in the phosphorylation of the STAT3 protein, STAT5 protein, and JAK2 protein by reducing p-STAT5, p-JAK2, and p-STAT3 protein intensities compared to untreated cells. The results also showed that TQ decreased JAK2, STAT3, and STAT5 protein levels, as illustrated in Table 2 and Figure 5. The findings are in agreement with previous findings in which the phosphorylation of JAK2, STAT3, and STAT5 was decreased in K562 cells after Dehydrocostus lactone treatment, the phosphorylation of STAT3 was reduced after treatment with a JAK inhibitor (AG490), and the phosphorylation of STAT5 was decreased with imatinib mesylate treatment in K562 cells [46]. Additionally, TQ inhibited STAT3 phosphorylation in Multiple myeloma cells [47] and MV4-11 leukemia cells [27]. Moreover, in our previously published report, we indicated that TQ had reduced the phosphorylation and the protein levels of JAK2, STAT3, and STAT5 protein intensities and phosphorylation in MV4-11 AML cells [48,49]. The findings of this study are also in agreement with previous findings in which TQ treatment reduced the phosphorylation of JAK2 and STAT3, which consequently led to enhanced apoptosis in SK-MEL-28 melanoma cells [50]. Moreover, TQ treatment has been found to induce apoptosis of B16-F10 melanoma cells and antitumor activity in a murine model of intracerebral melanoma through the inhibition of STAT3 phosphorylation [51]. TQ treatment has also decreased the phosphorylation of STAT3, which led to the inhibition of cell proliferation of PC3 human prostate cancer cells [52].

Previous studies have demonstrated that inhibiting JAK2/STAT3 and STAT5 signaling induces apoptosis in K562 cells and delays cell cycle progression [46]. This study evaluated the effect of hypomethylation-associated inhibition of JAK/STAT signaling induced by TQ on apoptosis and cell cycle progression in K562 cells. The results demonstrated that TQ enhanced apoptosis in K562 cells in a dose and time-dependent manner. TQ caused 91% total apoptosis following 72 h of treatment with 23 µM in comparison to 79% and 48% total apoptosis after treatment with 15 μM and 11 µM, respectively, at the same period of exposure. Additionally, the percentage of total apoptosis was increased by extending the incubation period with 23 µM TQ from 53.7% to 97% (Figure 1A,B). In agreement with our findings, TQ has been shown to induce apoptosis due to the reduction of DNA methylation in MV4-11 leukemia cells [27,36]. The findings of this study are also supported by other studies in which TQ enhanced apoptosis in C91, HuT-102, CEM, Jurkat human T-cell acute lymphoblastic leukemia cells [25,26,31], MV4-11 AML cells [27], and HL60 AML cells [26,40,53].

Previous studies have indicated that TQ induces cell cycle arrest in different cancers, including leukemia [25,54]. In addition, TQ has been reported to induce cell cycle arrest at the G0 phase in HuT102, C91, CEM, and Jurkat acute lymphoblastic leukemia cells after 48 h of treatment [25,31]. TQ has also been reported to arrest the cell cycle at the G0 phase in breast cancer MCF7 cells after 72 h of treatment [55]. Similarly, our findings indicated that TQ significantly arrested the K562 cell cycle at the G2/M phase and decreased the cells in G0/G1 in a concentration and time-related manner after 24 h and 48 h of treatment (Figure 4A). The highest significant accumulation of the cells in G2/M (42.84%) was attained at 23 μM concentration after 48 h of TQ treatment compared to 26.26% in untreated cells. Extending the treatment duration to 72 h caused accumulation of the cells at the G2/M and G0 phases and decreased the G0/G1 population (Figure 2A,B). The arrested cells at the G0 phase were increased from 2.91% in the untreated cells to 9.23% by increasing TQ concentration, which indicates that the cells started to undergo apoptosis after extending the period of treatment. In agreement with our results, TQ induced cell cycle arrest at the G0 phase in breast cancer MCF7 cells after 72 h of treatment [55].

Taken together, TQ induced anti-leukemia activities in K562 CML cells. TQ-induced apoptosis and cell cycle arrest were a consequence of generating a balanced expression of the genes that regulate the epigenetic mechanism, which resulted in hypomethylation and re-expression of *SHP-1*, a JAK/STAT negative regulator gene, which in turn led to inhibition of JAK/STAT signaling.

## 4. Materials and Methods

### 4.1. Cell Culture

Human CML cells (K562) with *BCR-ABL* mutation were obtained from the ATCC (MD, USA). RPMI medium, penicillin/streptomycin (P/S) solution and fetal bovine serum (FBS) were acquired from Elabscience company (Wuhan, China). Complete media (RPMI media) supplemented by P/S (1%) and FBS (10%) was used to sustain K562 cells at 37 °C in a humidified incubator provided with 5% CO_2_. The cells were cultured at a number of 5 × 10^4^ cells/mL and incubated until they reached 80% confluence.

### 4.2. Preparation of 5-Azacytidine and Thymoquinone Solutions

TQ (purity > 98%) was provided by Sigma-Aldrich company (Sigma-Aldrich, St. Louis, MO, USA). Dimethyl sulfoxide (DMSO) (2%) was used to dissolve TQ at a 30 mM stock concentration. TQ stock solution was then stored at −80 °C. At the time of treatment, working concentrations of TQ were made with RPMI media. The demethylating drug, 5-Azacytidine, was provided by Beijing Solarbio company (Beijing, China). A concentration of 10 mM 5-Aza stock solution was prepared with DMSO (2%) and then stored at −80 °C. A concentration of 2.3 µM 5-Aza working solution was prepared with RPMI medium at the time of treatment.

### 4.3. Analysis of Apoptosis by Flow Cytometer

The apoptosis detection kit (Annexin V/PI) provided by Nacalai-Tesque (Kyoto, JAPAN) was utilized for evaluating the K562 cell’s apoptotic events. TQ was added to the cells at their respective IC50 Concentrations, 23, 15, and 11 μM, and incubated for 24, 48, and 72 h, respectively. Both treated and untreated cells were washed with phosphate-buffered solution (PBS) twice. The cells were then resuspended into Annexin V-binding buffer with a final concentration of 1 × 10^6^ cells/mL. Then, 5 μL of the PI solution and 5 μL Annexin V-FITC Conjugate were added to the cell suspension (100 μL) and protected from light for 15 min at RT. Annexin V-binding buffer (400 μL) was then added to the cell solution. Cy-toFLEX flow cytometer (Beckman Coulter, Brea, CA, USA) was used for the measurements of apoptosis. The Software, Cyt-Expert for Cyto-FLEX (Beckman Coulter, Brea, CA, USA) was used to acquire and analyze data from 10,000 events.

### 4.4. Cell Cycle Analysis by PI Staining Assay

The cell cycle analyses were done using the Fluorometric-red-cell cycle kit from Elabscience company (Wuhan, China). TQ treatment of K562 cells was conducted as in the apoptosis assay. The cells were then centrifuged and resuspended into PBS to prepare 5 × 10^5^ cells/tube and further resuspended into 300 μL of PBS. After that, 1.2 mL of absolute iced ethanol was added. To fix the cells, they were incubated overnight on ice. After fixation, cells were centrifuged, resuspended with PBS (1 mL), and maintained at room temperature for a period of 15 min. The cell solutions were centrifuged and resuspended with RNase-A reagent (100 μL). After that, the cells were incubated in a 37 °C water bath for 30 min. Subsequently, the cells were mixed with 400 μL of PI staining and maintained at 4 °C for 30 min. Then, a flow cytometer (Beckman Coulter, Brea, CA, USA) and ModFit LT 4.1 software (Beckman Coulter) were used to analyze the cell cycle progression.

### 4.5. RT-qPCR for the Analysis of Gene Expression

The untreated cells and TQ-treated cells (15 µM for a period of 48 h) were used to extract total RNA using an RNA extraction kit (ReliaPrep-RNA-Cell-Miniprep System) (Promega, Madison, WI, USA), depending on the manufacturer’s guidelines. Evaluation of the extracted RNA’s concentration and purity was performed using Implen Nanodrop-photometer (Weslake-Village, CA, USA). The Go-Taq 2-Step RT-qPCR kit (Promega, Madison, WI, USA) was utilized to synthesize cDNA. The expression of target genes was evaluated by SYBR Green-based kit (SYBR Green-based Go-Taq 2-Step RT-qPCR) (Promega, Madison, WI, USA), depending on the manufacturer’s guidelines. The StepOnePlusTM thermocycler from Applied Biosystems (Foster City, CA, USA) was used to perform the RT- qPCR. The data were analyzed using StepOne software version 2.3 (New York, NY, USA) after normalization with *β-actin* levels. The fold changes in expression (2^−∆∆Ct^) were determined. The primer sequences for the target genes were listed in our previously published article [36].

### 4.6. Extraction and Bisulfite Treatment of DNA

K562 cells were sustained for a period of 48 h with TQ (15 µM). As a positive control, cells were treated for 48 h with 2.3 µM of 5-Azacitydine. Completely methylated and unmethylated human genomic DNA (Bisulfite-treated) and completely unmethylated DNA (Bisulfite-untreated) were purchased from Qiagen (Qiagen, Hilden, Germany) and used as controls. The Wizard-Genomic DNA Purification assay (Promega, Madison, WI, USA) was used for DNA extraction before and after treating the cells, depending on the manufacturer’s guidelines. Using an Implen Nanodrop-photometer (Weslake-Village, CA, USA), the extracted DNA concentration and purity were determined. The EpiTect Bisulfite Kit from Qiagen (Hilden, Germany) was used to treat 2 μg of the extracted DNA with bisulfite, following the manufacturer’s directions.

### 4.7. Designing Primers for Pyrosequencing Analysis

The methylation of the bisulfite treated DNA was quantified using pyrosequencing of the *SHP-1* promoter 2 region. The CpGs islands were selected in the promotor region, flanking the 5′-untranslated region (5′UTR). We studied 4 CpG sites in the selected CpG islands. Using Py-roMark Assay Design Software v2.0 from Qiagen (Hilden, Germany), the primers for pyrosequencing analysis and PCR amplification were designed. The primers were synthesized by Qiagen PyroMark-Custom Assay (Hilden, Germany) for pyrosequencing analysis and PCR amplification. To allow for biotinylated PCR products, biotinylated-reverse primers were prepared. The analyzed promoter region and the sequences of the corresponding primers used for PCR amplification and pyrosequencing analysis were listed in our previously published article [36].

### 4.8. Pyrosequencing Analysis

A PyroMark PCR kit from Qiagen (Hilden, Germany) was used to perform the PCR reactions. For each PCR reaction, 20 ng of bisulfite-converted DNA was amplified in 50 µL of reaction mixture containing: 2× PyroMark PCR Master Mix (25 μL) (containing dNTPs, HotStarTaq DNA Polymerase, and 1× PyroMark PCR Buffer), 10× CoralLoad concentrate (5 μL), forward primer (0.2 µM) (1 μL), and 5′-biotinylated reverse primer (0.2 µM) (1 μL). The cycling conditions were as follows: 15 min at 95 °C for PCR activation, then 45 denaturation cycles for 30 s at 95 °C, an annealing step for 30 s at 56 °C, an extension step for 30 s at 72 °C, followed by a step of final extension for 10 min at 72 °C using a thermal cycler (Applied Biosystems, USA). The products of the PCR reactions were checked by gel electrophoresis in 1.2% agarose gels and visualized using Florosafe DNA staining (1st base Laboratories, Selangor, Malaysia). Images of the DNA bands were captured under ultraviolet transillumination using Syngene G-BOX (G: BOX-CHEMI-XL1.4, USA). The PCR products were sent to the Institute for Research in Molecular Medicine (INFORM) at Universiti Sains Malaysia (USM) (INFORM, USM, Kelantan, Malaysia) for pyrosequencing analysis. The PyroMark Q96 System (Qiagen, Hilden, Germany) and Pyro Gold reagents (Qiagen, Hilden, Germany) were used for pyrosequencing analysis. Pyrosequencing results are the mean methylation percentages of all observed CpG sites and were calculated for the *SHP-1* gene.

### 4.9. Preparation of Cellular Lysates

For 48 h, K562 cells were exposed to TQ (15 M). Cellular lysates were prepared using RIPA buffer (Nacalai-Tesque, Kyoto, Japan). The concentrations of the protein were quantified by the Bradford protein assay [56], using Coomassie brilliant blue (CBB) solution and bovine serum albumin (BSA).

### 4.10. Evaluation of Protein Intensities by Jess-Simple Western Analysis

The expression of proteins was evaluated according to a previously published article [48] using the capillary-based Protein-Simple-Jess system (Jess, CA, USA). The primary antibodies utilized for protein analyses were listed in our previously published article [48].

### 4.11. Statistical Analysis

Version 25 of the Statistical Package for the Social Sciences (SPSS) (SPSS, Chicago, IL, USA) was used to analyze the data. The Wilcoxon signed-rank test was used to compare TQ-treated K562 cells to control cells. Repeated-measure of ANOVA along with Bonferroni correction was used to compare data between the groups, and *p*-values of 0.05, 0.01, and 0.001 were considered statistically significant.

## 5. Conclusions

TQs anti-leukemia activities were evaluated in K562 CML cells. TQ potentially induced apoptosis and cell cycle arrest of K562 CML cells at the G2/M and G0 phases. These anti-leukemic activities of TQ could be attributed to the creation of a balanced expression of the genes that regulate the epigenetic mechanism; this is by downregulation of the expression of *DNMT1*, *DNMT3A*, and *DNMT3B* and upregulation of the expression of *WT1* and *TET2*, resulting in hypomethylation and re-expression of *SHP-1*. As a consequence, TQ decreased the protein and phosphorylation of JAK2, STAT3, and STAT5. These findings indicate that TQ, by reducing DNA methylation of genes that negatively regulate the JAK/STAT pathway, may represent a promising therapeutic alternative for the treatment of patients suffering from CML in the future.

## Figures and Tables

**Figure 1 pharmaceuticals-16-00884-f001:**
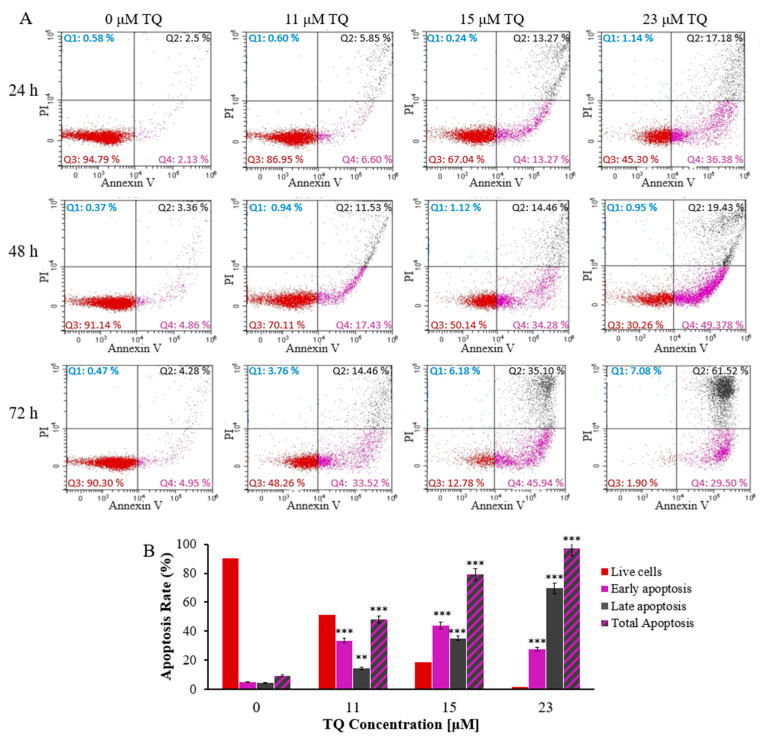
TQs time and dose-related apoptotic effects on K562 cells. (**A**) IC_50_ concentrations of TQ were used to treat the cells at the designated times. The flow cytometry and Annexin V-FITC/PI kit were used to assess events of apoptosis in K562 cells. (**B**) The percentages of K562 cells after treatment for 72 h are shown in the bar graph. Repeated-measures ANOVA was performed. Results are represented in Mean ± Standard Deviation (*n* = 3), in which ** *p* < 0.01 and *** *p* < 0.001 are significant in comparison to untreated cells.

**Figure 2 pharmaceuticals-16-00884-f002:**
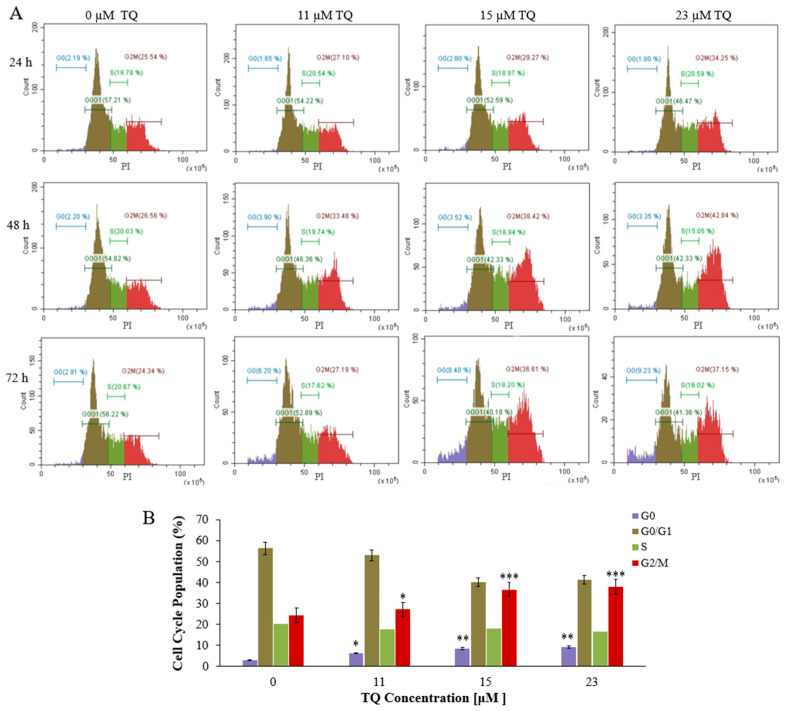
Histograms illustrating the analysis of the cell cycle in K562 CML cells. (**A**) K562 cells were exposed to the IC50s of TQ; 23 μM, 15 μM, and 11 μM, then stained with PI and analyzed by flow cytometry after 24, 48, and 72 h of incubation. After the treatment for 24 h and 48 h, TQ caused a significant elevation in the number of cells at the G2/M phase in K562 cells in a time and concentration-dependent manner. After 72 h of treatment, TQ significantly increased the G2/M and G0 populations. (**B**) The cell cycle distribution of K562 cells is illustrated by the bar graph after treatment with the IC50s of TQ for 72 h. Repeated-measures ANOVA was performed. Results are represented in Mean ± Standard Deviation (*n* = 3), in which * *p* < 0.05, ** *p* < 0.01, and *** *p* < 0.001 are significant in comparison to untreated cells.

**Figure 3 pharmaceuticals-16-00884-f003:**
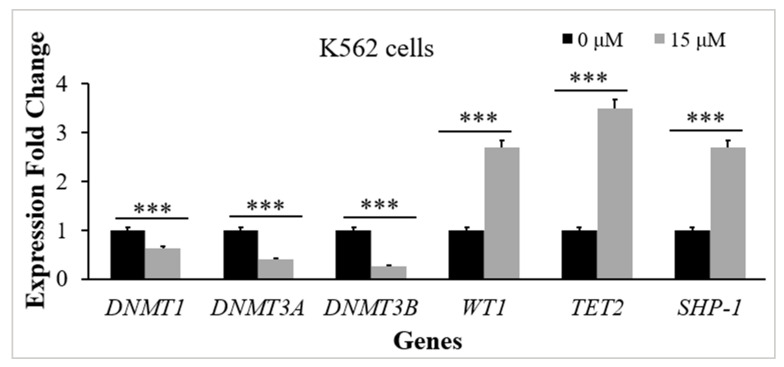
TQs influence on the target genes in K562 CML cells. K562 cells were incubated with 15 μM of TQ for a period of 48 h. Target genes were examined by RT-qPCR, and the bar graph demonstrates that TQ downregulated the *DNMT1* gene, *DNMT3A* gene, and *DNMT3B* gene expression and upregulated *TET2* gene, *WT1* gene, and *SHP-1* gene expression. Data were analyzed using the Wilcoxon signed-rank test. Results are expressed in median (IqR) (*n* = 3), in which *** *p* < 0.001 is statistically significant in comparison to controls.

**Figure 4 pharmaceuticals-16-00884-f004:**
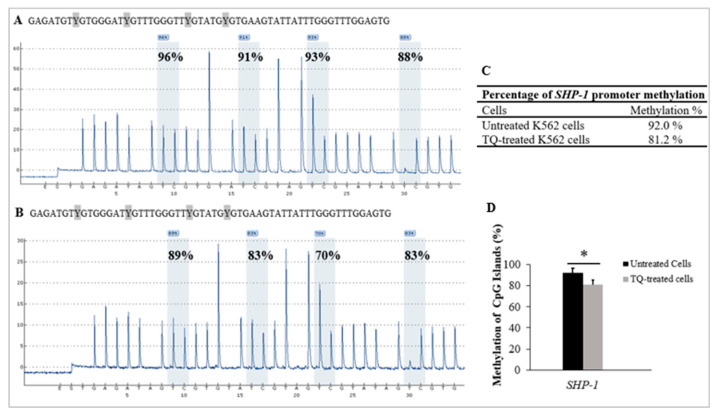
TQs effect on *SHP-1* promoter methylation in K562 cells. Pyrograms of pyrosequencing analysis of *SHP-1* promoter methylation of (**A**) the untreated control K562 cells and in (**B**) TQ treated K562 cells. Methylation levels at CpG sites are displayed as a percentage above each ‘‘C’’ base. (**C**) The table shows the percentages of methylation of the *SHP-1* promoter. (**D**) *SHP-1* hypomethylation in K562 cells by TQ treatment is illustrated by the bar graph. The repeated-measures ANOVA was conducted. Results are expressed as mean ± SD, in which * *p* < 0.05 is statistically significant compared to controls.

**Figure 5 pharmaceuticals-16-00884-f005:**
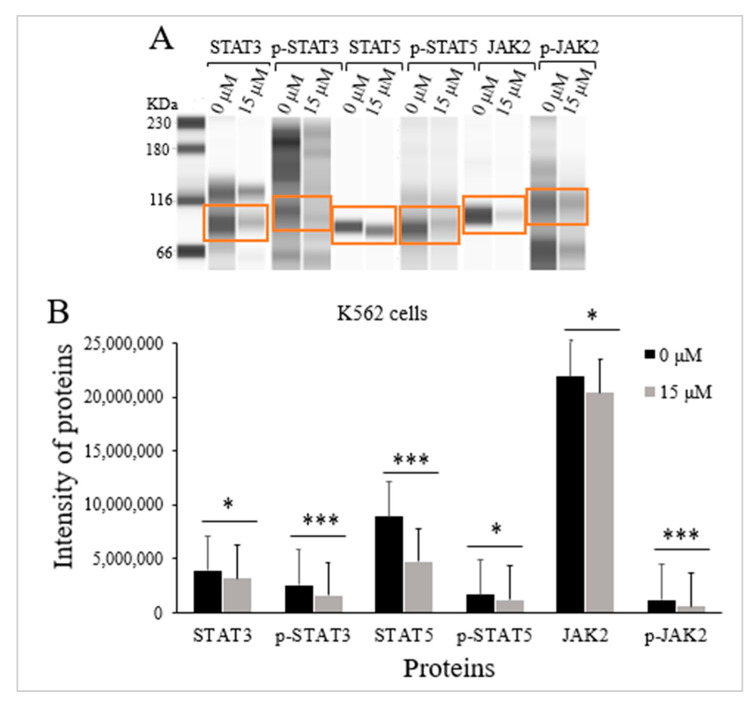
TQs effect on JAK/STAT signaling activation state in K562 cells. A concentration of 15 μM of TQ was used to treat K562 cells for 48 h. Jess-simple Western analyses were used in determining the intensities of target proteins. (**A**) The target proteins’ images using Jess-simple Western analysis for TQ-treated and untreated cells. (**B**) The decrease in the intensities of target proteins after TQ treatment of K562 cells is demonstrated by the bar graph. Data were analyzed using the Wilcoxon signed-rank test. Results are expressed in median (IqR) (*n* = 3), in which * *p* < 0.05 and *** *p* < 0.001 are statistically significant in comparison to controls.

**Table 1 pharmaceuticals-16-00884-t001:** The percentage of CpG sites methylated at the *SHP-1* promoter.

	CpG.1	CpG.2	CpG.3	CpG.4	Mean ± Standard Deviation
Untreated-K562 cells	96	91	93	88	92.0 ± 22.8
TQ-treated K562 cells	89	83	70	83	81.2 ± 11.0
5-Aza-treated K562 cells	70	58	68	63	64.7 ± 12.1
Unmethylated DNA control	6	0	0	0	1.5 ± 0.4
Methylated DNA control	95	88	100	84	91.8 ± 15.7
Unmethylated Bisufite-unconverted DNA control	0	0	0	0	0

**Table 2 pharmaceuticals-16-00884-t002:** The median levels of protein expression in K562 CML cells.

Proteins	M.Ws (kDa)	Proteins’ Intensities Medians (IqR)		*p*-Value ^a^
		Untreated K562 Cells	TQ-treated K562 Cells	
STAT3	80–90	3,891,993 (312,350)	3,153,914 (262,434	<0.05
p-STAT3	95	2,559,531 (254,110)	1,576,590 (131,188)	<0.001
STAT5	91	8,924,147 (295,549)	4,762,098 (192,331)	<0.001
p-STAT5	92	1,648,065 (170,566)	1,221,741 (154,917)	<0.05
JAK2	130	21,887,799 (501,710)	20,385,796 (423,651)	<0.05
p-JAK2	130	1,225,525 (103,221)	601,966 (91,423)	<0.001

^a^ The Wilcoxon signed-rank test was implemented. Medians of the proteins intensities were statistically significant compared to untreated cells.

## Data Availability

Data is contained within the article.

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
