# Peer review of "Thymoquinone Enhances Apoptosis of K562 Chronic Myeloid Leukemia Cells through Hypomethylation of SHP-1 and Inhibition of JAK/STAT Signaling Pathway"

_pharmaceuticals, 2023, doi:10.3390/ph16060884_

Round 1

Reviewer 1 Report

The introduction can be improved. Please, describe more deeply thymoquinone and its use in other cancer types.

Please describe the results section further and add some references for comparing the results with previous studies.

In the discussion section, the authors should propose mechanisms for thymoquinone-induced anticancer effects.

The authors might add an integrative figure to visualize better its results.

Author Response

Comments and Suggestions for Authors

Comment 1

The introduction can be improved. Please, describe more deeply thymoquinone and its use in other cancer types.

Authors Response:

Dear reviewer, thank you for your constructive comments, and please be informed that thymoquinone and its effects on other cancers have been added to the introduction as shown in the manuscript track changes (lines 72-77).

Comment 2

Please describe the results section further and add some references for comparing the results with previous studies.

Authors Response:

 Dear reviewer, please be informed that more references for comparing the results with previous studies have been added to the discussion as shown in the manuscript track changes (lines 255-261, lines 271-275, lines 277-280).

Comment 3

In the discussion section, the authors should propose mechanisms for thymoquinone-induced anticancer effects. 

Authors Response:

Dear reviewer, please be informed that the last paragraph in the discussion indicates and proposes the mechanisms for thymoquinone-induced anticancer effects, (please refer to the paragraph ij the revised manuscript between lines 297-300)

Comment 4

The authors might add an integrative figure to visualize better its results.

Authors Response:

Dear reviewer, thank you for your valued suggestion, and a graphical abstract has been created and uploaded as a ppt format which summarizes the results.

Reviewer 2 Report

In this paper, the authors have investigated the “in vitro” anti-leukemia activities of thymoquinone on K562 cells. The results reported, while not new since the same effects have been previously demonstrated in other cell lines, may be of interest. However, in my opinion, several points in the manuscript are unclear and need explanation. The main points are as follow:

-        The significance of the G0 phase of the cell cycle is unclear. Is it different from G0/G1 Does it refer to apoptosis? Is it the subG1?

-        The table 1 is not clear. Is it necessary? What does it add to the results already shown in Figure 4? Explain in the results section the different treatments shown in the last column on the left. The numbers in the right last column are also unclear.

-        Is section 2.5 necessary? It does not add anything to section 2.3.

-        Is Table 2 necessary? It adds nothing compared to Figure 5.

Two main limitations of this study that should be discussed in more detail are:

1) Because of the very small decrease in protein expression shown in Figure 5, are these effects physiologically significant, particularly in vivo? 

2) Normal cells (e.g., lymphocytes) have not been tested for the safe effects of thymoquinone. Is this molecule toxic to normal cells?

Author Response

Comments and Suggestions for Authors

In this paper, the authors have investigated the “in vitro” anti-leukemia activities of thymoquinone on K562 cells. The results reported, while not new since the same effects have been previously demonstrated in other cell lines, may be of interest. However, in my opinion, several points in the manuscript are unclear and need explanation. The main points are as follow:

Authors Response:

Dear reviewer please be informed that our study evaluates the anticancer activities of TQ on K562 cells which represent chronic myeloid leukemia (CML), which are different from MV4-11 cells which represent M5 subtype of AML.

Comment 1

The significance of the G0 phase of the cell cycle is unclear. Is it different from G0/G1 Does it refer to apoptosis? Is it the subG1? 

Authors Response:

Thank you for your valued comments, and please be informed that G0 phase is similar to subG1 and different from G0/G1. The G0 phase, also known as the resting phase, which indicates a metabolic arrest and apoptosis initiation.

The accumulation of the cells at G0 phase after 72 hours of treatment indicates that the cells start to undergo apoptosis after extending the period of treatment. Similar clarification was added to the discussion (lines 293-294).

Comment 2

The table 1 is not clear. Is it necessary? What does it add to the results already shown in Figure 4? Explain in the results section the different treatments shown in the last column on the left. The numbers in the right last column are also unclear. 

Authors Response:

Table 1 is necessary because it shows the results of pyrosequencing for the DNA controls which are important for deciding if the DNA is methylated or not. Table 1 also shows the results for the positive control (5-Azacitydine, hypomethylating agent). Figure 4 only shows the results before and after treatment with thymoquinone. The numbers in the right last column were mistakenly written in the first 3 cells of the column and were corrected as shown in the revised manuscript.

Comment 3

Is section 2.5 necessary? It does not add anything to section 2.3.

Authors Response:

Dear reviewer, section 2.5 is necessary and demonstrate that Thymoquinone enhances SHP-1 expression after its hypomethylation as demonstrated in section 2.4. Therefore, section 2.5 is not related section 2.4, nor to section 2.3 that shows the balanced expression of the genes that regulate DNA methylation.

Comment 4

Is Table 2 necessary? It adds nothing compared to Figure 5.

Authors Response:

Table 2 is necessary because it shows the intensities of the proteins and the molecular weights of the tested proteins at which they were detected during the analysis, which are not shown in Figure 5.

Comment 5

Two main limitations of this study that should be discussed in more detail are:

  • Because of the very small decrease in protein expression shown in Figure 5, are these effects physiologically significant, particularly in vivo?  

Authors Response:

Dear reviewer, thank you for your comments, please notice that the decrease in all protein expressions is statistically significant and the physiological significant was demonstrated by the apoptotic enhancement and cell cycle arrest which was highlighted in the last paragraph of the discussion and in the conclusion.

Currently our team is working on the in-vivo effect of thymoquinone to further identify thymoquinone-induced inhibition of JAK/STAT and other signaling pathways on apoptosis in a mice model with leukemia. The experiments are under investigation and the results will be published later.

  • Normal cells (e.g., lymphocytes) have not been tested for the safe effects of thymoquinone. Is this molecule toxic to normal cells?

Authors Response:  

The present study focused on the antileukemia effect of thymoquinone on K562 chronic myeloid leukemia cells. However, the effect of thymoquinone on activated peripheral blood mononuclear cells (PBMC) non-tumor cells had been investigated previously and the results showed that these cells were more resistant to thymoquinone treatment compared to cancer cells, suggesting that thymoquinone is not deleterious on non-malignant cells, please refer to; Dergarabetian, E. M., Ghattass, K. I., El-Sitt, S. B., Al-Mismar, R. M., El-Baba, C. O., Itani, W. S., ... & Gali-Muhtasib, H. U. (2013). Thymoquinone induces apoptosis in malignant T-cells via generation of ROS. Frontiers in bioscience (Elite edition)5, 706-719.

Reviewer 3 Report

General comments:

The authors report that TQ promotes apoptosis and cell cycle arrest in CML cells by inhibiting JAK/STAT signaling via restoration of the expression of JAK/STAT-negative regulator genes.

Major comments:

1. The evidence of apoptosis is only supported by annexin V/PI data (Fig. 1). The caspase signaling needs to be further assessed. At least the caspase 3 need to evaluate for this study.

Minor comments:

1. Fig 2B: the first “G0” in should be subG1. Please correct the figure label.

Author Response

Comments and Suggestions for Authors

General comments:

The authors report that TQ promotes apoptosis and cell cycle arrest in CML cells by inhibiting JAK/STAT signaling via restoration of the expression of JAK/STAT-negative regulator genes. 

Major comments:

  1. The evidence of apoptosis is only supported by annexin V/PI data (Fig. 1). The caspase signaling needs to be further assessed. At least the caspase 3 need to evaluate for this study.

Authors Response:

Many thanks to the reviewer, and with respect to your suggestion, please consider that flowcytometry analysis using annexin V/PI is the gold standard method to evaluate apoptosis. Therefore, the present study has provided strong evidence of TQ-enhanced apoptosis.

Minor comments:

  1. Fig 2B: the first “G0” in should be subG1. Please correct the figure label.

Dear reviewer, please be informed that G0 phase is similar to subG1. Fig 1B were acquired from the flowcytometry software with G0 instead of subG1. Therefore, it would be more consistent if this phase is identified by G0 instead of subG1 in Fig 2B label.

Round 2

Reviewer 2 Report

Although the manuscript remains verbose and repetitive with respect to the information it contains, I believe it can be published in its current form

Reviewer 3 Report

All reviewer's concerns were well-responded.